# Biomarkers of Myocardial Injury and Remodeling in Heart Failure

**DOI:** 10.3390/jpm12050799

**Published:** 2022-05-16

**Authors:** Barbara Ponikowska, Gracjan Iwanek, Agata Zdanowicz, Szymon Urban, Robert Zymliński, Piotr Ponikowski, Jan Biegus

**Affiliations:** 1Student Scientific Organization, Wroclaw Medical University, 50-556 Wroclaw, Poland; 2Institute of Heart Diseases, Wroclaw Medical University, 50-556 Wroclaw, Poland; agatazdanowicz@gmail.com (A.Z.); s.urban@umw.edu.pl (S.U.); robert.zymlinski@umw.edu.pl (R.Z.); piotrponikowski@4wsk.pl (P.P.); janbiegus@gmail.com (J.B.)

**Keywords:** heart failure, biomarkers, remodeling, heart injury

## Abstract

With its complicated pathophysiology, high incidence and prevalence, heart failure remains a major public concern. In hopes of improving diagnosis, treatment and prognosis, the utility of many different biomarkers is researched vigorously around the world. In this review, biomarkers of myocardial remodeling and fibrosis (galectin-3, soluble isoform of suppression of tumorigenicity 2, matrix metalloproteinases, osteopontin, interleukin-6, syndecan-4, myostatin, procollagen type I C-terminal propeptide, procollagen type III N-terminal propeptide, vascular endothelial growth factor, nitric oxidase synthetase and asymmetric dimethylarginine), myocyte injury (heart-type fatty acid-binding protein, glutathione S-transferase P1 and heat shock protein 60), as well as iron metabolism (ferritin, transferrin saturation, soluble transferrin receptor and hepcidin), are considered in terms of possible clinical applicability and significance. Our short review consists of a summary of the aforementioned cardiovascular biomarkers’ clinical relevance and perspectives.

## 1. Introduction

Heart failure (HF) is a syndrome with a complex and not fully recognized pathophysiology. In recent decades, our understanding of the key mechanisms involved in HF development and progression has evolved from a “simple” hemodynamic problem through a neurohormonal and proinflammatory-driven syndrome, to a complex multiorgan dysfunction accompanied by inadequate energy handling. Moreover, the pathophysiological considerations may be led at completely different levels, from molecular or cellular up to hemodynamic and clinical, complicating the syndrome’s interpretation and understanding.

Nowadays, the clinical approach mobilizes researchers to search for novel cardiovascular biomarkers that could present the value in risk stratification, early detection and therapeutic management (both as targets and clinical response quantifiers). As an example, natriuretic peptides (NPs) should be taken into consideration. N-terminal pro B natriuretic peptide (NTproBNP) and Midregional-pro A natriuretic peptide (MRproANP) are released into the circulation in response to pressure stretching of cardiomyocytes [1]. The value of NTproBNP as a diagnostic tool is reflected in the ESC guidelines for the diagnosis and treatment of HF [2]. Additionally, explorers of neprilysin inhibitors marked natriuretic peptides as targets for pharmacological therapy.

Nevertheless, the dysfunction of the cardiovascular system itself is believed to be central to the cascades of other mechanisms. Cardiac (and probably vascular) injury and remodeling are one of the elements that is crucial for the development of cardiovascular dysfunction in HF. On the other hand, iron deficiency, which is a therapeutic target in HF, may be related to energy debt at the cellular level. However, despite persistent efforts and numerous basic research works showing the pathophysiological significance of many biomolecules involved in disease development, only a few of them have real clinical significance and therefore utility. In this article, we review the major biomarkers involved in the processes of myocardial injury, remodeling and iron metabolism, and we try to discuss their value from a clinical perspective.

## 2. Biomarkers of Myocardial Remodeling

### 2.1. Cardiovascular Extracellular Matrix

The cardiac extracellular matrix (ECM) is a highly dynamic acellular network that surrounds all cardiomyocytes. The ECM provides a background for the main homeostatic reactions related to cardiac remodeling, fibrosis and subsequent heart failure pathogenesis [3]. This is why the ECM is the area of interest when searching for new cardiovascular biomarkers (Table 1). Here, we focus only on cardiac extracellular biomarkers that can modulate the cardiomyocyte biology in health and disease. However, from the clinical point of view, it is important to understand that the extracellular matrix of peripheral tissues may be equally important. The best examples of this phenomenon are other (than heart) organ dysfunctions observed in heart failure, as well as the key involvement of the interstitial compartment in the development of congestion in acute heart failure (AHF).

#### 2.1.1. Galectin-3

Galectin-3 is a soluble β-galactoside-binding protein that modulates cell adhesion processes by interacting with ECM proteins. Galectin-3 plays a key role in heart remodeling mechanisms, especially cardiac hypertrophy, and fibrosis [4,5]. Galectin-3 is associated with an increased HF risk in the general population, higher cardiovascular and all-cause mortality [6]. In several research works, the neutralization of Galectin-3 brought about limitations of pro-fibrotic remodeling processes [7,8].

#### 2.1.2. The Soluble Isoform of Suppression of Tumorigenicity 2

The soluble isoform of suppression of tumorigenicity 2 (sST2), structurally similar to interleukin-1, is a known inflammatory and fibrotic agent. Its receptors are located, among others, on cardiomyocytes and cardiac fibroblasts. As a biomarker, it is useful for HF risk stratification, independently of natriuretic peptides and highly sensitive troponins [9,10]. High serum levels of sST2 were linked to a higher risk of incident HF and cardiovascular death [11]. The sST2-guided therapy seems to be a promising clinical tool and is under investigation in an ongoing clinical trial in a group of patients suffering from AHF [12].

#### 2.1.3. Matrix Metalloproteinases

Matrix metalloproteinases (MMP) are a family of 25 proteolytic enzymes involved in post-ischemic ECM turnover, an important factor in heart failure development [13]. Multiple research projects established the relation between MMP-9 and post-myocardial infarction (MI) heart failure and mortality prediction [14]. In addition, MMPs, with their specific tissue inhibitors (TIMPs), have been investigated as therapeutic targets [15]. This relation is widely described as an MMP/TIMP system, in which changes in the protease/antiprotease balance are important HF predictors [16].

#### 2.1.4. Osteopontin

Osteopontin (OPN) is an inter-ECM signal-transducing protein expressed in response to biomechanical stress [17]. OPN synthesis is upregulated in the post-MI heart [18] and severe HF due to dilated cardiomyopathy [19]. OPN plasma levels may be useful to assess HF severity [20]. OPN protein may be considered as an HF severity marker, while OPN mRNA is associated with reverse cardiac remodeling [21]. Osteopontin (OPN) is an inter-ECM signal-transducing protein expressed in response to biomechanical stress [17]. OPN synthesis is upregulated in the post-MI heart [18] and severe HF due to dilated cardiomyopathy [19]. OPN plasma levels may be useful to assess HF severity [20]. The OPN protein may be considered an HF severity marker, while the OPN mRNA is associated with reverse cardiac remodeling [21]. Inhibition of OPN-mediated signaling is shown to be an effective approach for preventing cardiac hypertrophy and fibrosis, improving cardiac function and reversing pressure overload-induced heart failure [22].

#### 2.1.5. Interleukin-6

Interleukin-6 (IL-6) is an important inflammatory mediator that is secreted to the circulatory system in response to infections and tissue injuries in the acute phases [23]. The BIOSTAT CHF (Biology Study to Tailored Treatment in Chronic Heart Failure) revealed that IL-6 independently predicted primary outcome (defined as the combined outcome of all-cause mortality and unscheduled hospitalization for HF in 2 years interval), all-cause mortality, cardiovascular and non-cardiovascular mortality [24]. Additionally, a study based on the PREVEND (Prevention of Renal and Vascular end-stage Disease) cohort presented that IL-6 was statistically significantly associated with the development of HFpEF (heart failure with preserved ejection fraction) [25]. High levels of both serum and urine IL-6 correspond to diuretic resistance, lower eGRF and increased renal tissue-level neurohormonal activation in HF patients [26].

#### 2.1.6. Syndecan-4

Syndecan-4 (Syn-4) is a transmembrane glycoprotein involved in signal transduction, resulting in tissue regeneration, angiogenesis and focal adhesion [27]. It is activated by growth factors and ECM proteins [28]. As a remodeling biomarker, it is significantly correlated with left ventricle geometrical parameters in chronic heart failure (CHF). Serum levels of Syn-4 were shown to be correlated with left ventricle dimensions [27,29,30].

#### 2.1.7. Myostatin

Myostatin (MSTN) is an ECM signaling molecule functionally considered a negative muscle mass regulator [31]. MSTN null post-MI mice model showed protective attributes of MSTN absence in cardiac remodeling understood as the greater recovery of EF (ejection fraction) and lesser collagen deposition [32]. MSTN serum levels showed predictive value for HF’s severity and clinical outcome. CHF patients present higher MSTN serum levels. Additionally, higher levels of serum MSTN in CHF patients are significantly related to a lower survival rate and a larger number of rehospitalizations [33].

### 2.2. Collagen Metabolism

Collagen metabolism is an important factor in the cardiac fibrosis process. The cardiac ECM contains mainly collagen type I and III. The synthesis of collagen is an organized, multi-level reaction. Intermediate products of this process can be quantified and used as remodeling and fibrosis biomarkers [34].

#### 2.2.1. Procollagen Type I C-Terminal Propeptide

Procollagen Type I C-terminal propeptide (PICP) serum levels correspond to the active synthesis of collagen type I. Elevated PICP serum levels were found in HF patients of hypertensive origin [35]. Additionally, PICP serum levels were correlated with ventricular hypertrophy and diastolic dysfunction in hypertensive patients [36]. Histopathologic analysis of myocardial samples obtained from patients with hypertrophic cardiomyopathy revealed the correlation between high serum PICP levels and proven cardiac fibrosis [37].

#### 2.2.2. Procollagen Type III N-Terminal Propeptide

Procollagen type III N-terminal propeptide (PIIINP) is secreted to the serum as a result of active collagen type III synthesis in cardiac fibrosis processes [38]. The MESA (Multi-Ethnic Study of Atherosclerosis) revealed the PIIINP value for early risk stratification for HFpEF (but not HF with reduced ejection fraction) in a population free of overt cardiovascular disease [39]. Furthermore, PIIINP has been shown as an independent all-cause and cardiovascular mortality predictor.

### 2.3. Vascular Endothelial Growth Factor

Vascular endothelial growth factor (VEGF) is a cytokine involved in cardiac remodeling throughout angiogenesis processes. There are five subtypes of VEGF: VEGF-A, VEGF-B, VEGF-C, VEGF-D, and placental growth factor (PIGF). VEGF-A and VEGF-B are responsible for neovascularization; their signaling path is transduced through VEGF type 1 and type 2 receptors (VEGFR-1, VEGFR-2). VEGF-C and VEGF-D bind to VEGF type 3 receptor (VEGFR-3) and promote lymphangiogenesis. VEGF subtypes are under investigation as novel heart failure biomarkers and potential therapeutic agents [40,41].

The PREHOSP-CHF (Development of Novel Biomarkers to Predict REHOSPitalization in Chronic Heart Failure) study demonstrated the value of VEGF-C and VEGF-D as CHF outcome predictors. VEGF-C serum levels were inversely associated with the composite endpoint, defined as cardiovascular death or HF rehospitalization and non-cardiovascular death. Serum levels of VEGF-D were positively associated with an increased risk of HF rehospitalizations [42]. Low levels of VEGF may reflect lymphatic dysfunction or insufficiency in HF that may correlate with peripheral tissue congestion, which should be distinguished from high intracardiac pressure markers, such as natriuretic peptides.

### 2.4. Nitric Oxidase Synthetases

Nitric oxidase synthetases (NOS) are known cardiovascular bioregulators that act as endothelium-derived relaxing factors. Inhibitors of NOS are enzymes responsible for regulating nitric oxide bioavailability in acute heart failure and cardiovascular remodeling [43].

#### Asymmetric Dimethylarginine

Asymmetric dimethylarginine (ADMA) is an endogenous NOS inhibitor that has been widely investigated as an HF risk stratification and outcome biomarker [44]. High serum concentrations of ADMA were found in patients that developed cardiogenic shock after acute myocardial infarction. ADMA serum levels were strong, independent predictors of 30-day mortality in this group of patients [45]. Higher serum concentrations of ADMA were significantly associated with all-cause mortality in patients with cardiovascular disease, especially in the non-diabetic group [46]. In HF patients, increased ADMA concentrations were related to reduced renal perfusion [47].

### 2.5. Clinical Perspectives

Extracellular matrix remodeling of the cardiovascular system is a fundamental process in heart failure pathophysiology. At the moment, however, most of the discussed biomarkers have limited clinical applicability and rather have significance for a better understanding of the pathways leading to myocardial dysfunction. The early trials with MMP inhibitors did not show unambiguous positive results. On the other hand, the markers of matrix remodeling and inflammation (such as IL-6) seem to be a very promising therapeutic target in both acute and chronic HF settings. Moreover, the increasing interest in lymphatic system involvement in AHF shows huge promise for the clinical utility of biomarkers of lymphatic dysfunction and failure (such as VEGF-C). Moreover, there are real clinical premises to believe that markers of lymphatic dysfunction (i.e., VEGF family members) may be used as a “perfect” reflection of peripheral congestion, as the natriuretic peptides that we use in clinical practice are actually markers of intracardiac pressure, rather than a marker of interstitial pressure.

## 3. Biomarkers of Myocyte Injury

Myocardial injury/insult is one of the key pathophysiological processes underlying the development and progression of heart failure (HF) syndrome. The death of cardiomyocytes, due to either apoptosis and/or necrosis, is a well-described feature of HF with an ominous clinical impact [42,43]. Thus, the identification of biomarkers that can reliably characterize this process and have an impact on clinical decision-making strategy has remained a challenging task in the last few decades.

### 3.1. High-Sensitivity Troponins

In contemporary clinical practice, high-sensitivity troponins (hs-Tn) are widely used as markers of myocardial injury/necrosis in chronic and acute settings of HF [48,49]. Recent advances in the troponin assays have led to the development of highly (or ultra)-sensitive assays able to detect TNI at much lower concentrations, which potentially offers a more precise tool to evaluate and monitor the magnitude of cardiac injury during the course of the disease. Elevated serum levels of hs-Tn have strong prognostic significance for adverse clinical outcomes in acute and chronic HF [49] and therefore their assessment is recommended for this purpose [2].

Notably, it was shown that elevated hs-TNI was present in the majority of patients admitted to the hospital for AHF [50] Additionally, the significant increase in the hs-Tn level during the hospital stay was shown to be a predictor of cardiovascular mortality [51]. The guidelines also recommend the assessment of serum troponins in acute AHF to exclude/confirm acute coronary syndrome as a cause of decompensation, or in patients with cancer to monitor the potential cardiotoxicity of anti-cancer therapies, as well as in the workup of suspected myocarditis [2]. However, there is still no troponin–based therapeutic algorithm for HF.

It is important to understand that there are numerous conditions (e.g., cerebrovascular diseases, sepsis, chronic kidney disease, connective tissue diseases, neoplasms, invasive medical procedures, strenuous physical exertion) in which hs-Tn are elevated and the clinical meaning of this elevation is difficult for unequivocal judgement [52]. Moreover, the exact mechanism of the TNI rise in non-cardiac disorders remains unelucidated, as several complementary mechanisms have been postulated to play a role, including myocardial injury, myocardial death, increased permeability of the myocardial cell membrane, etc.

However, the hs-TNI elevation in the AHF setting may be interpreted as evidence for at least myocardial (cardiovascular) injury related to each episode of heart failure decompensation [53]. The magnitude of the TNI rise may reflect the magnitude of insult for all vital organs (liver, kidney, lungs) related to the AHF episode itself and leading to multiorgan dysfunction in AHF. Below, we briefly present some other, lesser known biomarkers which characterize myocardial injury.

### 3.2. Fatty Acid-Binding Proteins

Fatty Acid-Binding Proteins are small lipid-binding proteins, found in many tissues of the body, notably in those with a high fatty acid metabolism (e.g., heart, kidneys, brain) [54]. Among different tissue-specific isoforms of FABP, the myocardial isoform (H-FABP) is present at high levels in cardiomyocytes, but can also be found in the skeletal muscles and kidneys, mammary glands and lungs. Due to the high expression in cardiomyocytes (mainly in the cytoplasm), its levels in the plasma rise rapidly to a detectable level within 20 min of myocyte injury, reaching peak levels after 3–4 h and returning back to baseline levels in 18–30 h [55]. Additionally, even minor cardiac damage can trigger plasma levels of H-FABP to rise [56]. Due to these properties, it would appear that H-FABP can become a reliable biomarker for diagnosis in a number of conditions involving myocyte injury, including acute coronary syndrome [54,56]. However, in the era of cardiac troponins, with their high negative predictive value and sensitivity, the added value of measuring H-FABP in addition to hsTn seems to be of less clinical applicability [54,56]. An overview of the clinical applications of H-FABP assessment is summarized and presented in Table 2. Of note, while serum levels of H-FABP rise, intracellular levels are reduced, which may further exacerbate chronic HF by means of disturbing cardiomyocytes’ homeostasis [56].

### 3.3. Glutathione S-Transferase P1

Glutathione S-transferase P1 (GSTP1), an isozyme of the glutathione S-transferase family, has a substantial role in the processes of regulating inflammation, cellular homeostasis and detoxification of reactive oxygen species (ROS) and maintenance of the cellular redox state [57]. In the heart of an HF patient, reactive oxygen species are more abundant due to hypertrophy and inflammation [58], and therefore, due to its role in the detoxification of ROS, the expression of GSTP1 rises in these patients. A study by Andrukhova et al. [57] has shown that GSTP1 is an independent, sensitive and specific predictor of LV function in HF, with higher specificity when compared to NTproBNP. It has been speculated that it is because GSTP1-dependent processes of inflammation affect the entire myocardium, independently of left ventricular mass and atrial dilation, which correlate with natriuretic peptide levels [57]. Regardless of these findings, more research on the relationship between GSTP1 and HF clinical characteristics would be necessary to consider serum GSTP1 a reliable biomarker useful in clinical practice.

### 3.4. Heat Shock Protein 60

Heat Shock Protein 60 (HSP60) is a mitochondrial protein present in the majority of cells, expressed as a result of various stimuli, such as infection, oxidative stress, anoxia and inflammation [59]. Detectable levels of serum HSP60 (sHSP60) can be spotted in healthy patients [60], but its levels are significantly elevated in patients with chronic HF, with a relationship with the advanced NYHA class [61], most likely as a result of its secretion from cardiac myocytes due to injury, as well as proinflammatory activation [59]. A study by Bonand et al. found a significant positive correlation between sHSP60 levels during an episode of HF decompensation and poor prognosis, including death and AHF readmission [59]. There is potential for the use of sHSP60 as a biomarker not only of myocyte injury but also the inflammatory and immune response in HF; however, a deeper understanding of the underlying physiological processes as well as the clinical application would be necessary.

### 3.5. Natriuretic Peptides

Despite the natriuretic peptides such as NTproBNP and MRproANP being released into the bloodstream rather as a result of intracardiac pressure overload than myocardial injury or remodeling, they present high additive diagnostic and prognostic value to the abovementioned molecules. The diagnostic significance of NPs is indisputable, but their possible utility is more advanced. In juxtaposition with biomarkers of myocardial injury and remodeling, they may be helpful in HF origin qualification and clinical responses to treatment evaluation [62]. Additionally, NPs have high importance in the search for novel cardiovascular biomarkers. Correlation of the potential biomarker with NPs (strong diagnostic tool and outcome predictor) in a multi-variable model approach is used to reveal its independence and new scope of utilization.

### 3.6. Clinical Perspectives

Myocardial injury and therefore dysfunction seem to be crucial for heart failure development and progression. However, apart from troponins and natriuretic peptides (which are part of the guideline-recommended assessment), most of the discussed biomarkers are not used in everyday clinical practice, but they are mostly applied in basic studies. Despite the fact that all presented molecules are classified as markers of myocardial injury, we need to understand that actually each of them represents a different pathophysiological process. This fact leads to the development of artificial intelligence-based algorithms that may look for specific constellations of the biomarkers (clusters)—different HF phenotypes, with different underlying pathophysiology and prognosis.

## 4. Biomarkers of Iron Metabolism

Iron deficiency (ID) is one of the most common comorbidities in HF, with a prevalence of 30–60% of patients with chronic HF, regardless of ejection fraction [63]. It tends to be more frequent in anemics, in women and in patients with more advanced HF (as evidenced by higher levels of natriuretic peptides and NYHA class) [63]. Recent reports also documented a high prevalence of ID in AHF, with a rate between 50 and 70% [63,64,65].

ID needs to be looked at independently of anemia as its deleterious consequences are far beyond the effects of low levels of hemoglobin and comprise mitochondrial dysfunction, impaired energetic processes, impaired reactive oxygen species and abnormalities in the immune response, to name a few [63,64,65]. Thus, the clinical problem of ID in HF has received considerable attention in recent times, also as a successful target for therapeutic intervention [63,64,65].

Definitions of ID typically differentiate between absolute and functional ID—the former reflects low (or even absent) iron stores in the body, whereas the latter characterizes a state with sufficient levels of stored iron with inadequate iron availability for all the processes utilizing iron in the body [64]. In functional ID, the low availability of iron can be a result of the downregulation of ferroportin 1, a protein that allows the outflow of iron from enterocytes, macrophages and hepatocytes into the extracellular space and into the bloodstream [64]. The causes underlying the high prevalence of ID in HF are multifactorial and not yet fully understood [63,64,65]. In HF specifically, ID can arise due to impaired absorption from the duodenum resulting from intestinal edema, inadequate iron dietary intake, multiple drug interactions, chronic gastrointestinal blood loss or other comorbidities (such as chronic kidney dysfunction) [63,64,65]. The pro-inflammatory and neurohormonal activation can also be factors leading to ID in HF [63,64,65].

Accurate assessment of iron metabolism and precise diagnosis of ID in HF is still a challenge. The bone marrow biopsy remains the gold standard for the best characterization of depleted iron stores in the body and diagnosis of absolute ID. However, this invasive procedure is now rarely used, being replaced by the assessment of several blood-borne biomarkers to depict iron status and diagnose ID. The iron biomarkers with established clinical recommendations in HF are briefly discussed below and presented in Table 3 [66].

### 4.1. Ferritin

Ferritin is an iron storage protein secreted by the liver and reticuloendothelial system; its levels in the serum correspond to its expression in the iron-storing tissues [63,67]. One significant disadvantage is that serum ferritin can be non-specifically elevated due to inflammation, which is present in HF but also in other chronic comorbidities [65,66].

### 4.2. Transferin Saturation

Transferrin saturation (TSAT), defined as the percentage of transferrin with iron bound to it, can be used as a measurement of the amount of iron that is available to supply metabolizing cells [67]. States of malnourishment and catabolic metabolism can lead to decreased serum transferrin disproportionate to serum iron, resulting in falsely elevated TSAT [68]. Together, TSAT and ferritin are commonly used to evaluate ID in patients. The 2021 ESC HF guidelines recommend the treatment of ID when the serum ferritin level is <100 μg/L, or when it is in the range between 100 and 299 μg/L when TSAT < 20% [67]. A recent study that investigated the utility of these two biomarkers in patients with HF and LVEF ≤ 45 % concluded that while TSAT was a useful diagnostic tool in identifying patients with ID, the value of ferritin was called into question [69].

### 4.3. Soluble Transferrin Receptor

Soluble Transferrin Receptor (sTfR) is a membrane receptor, present primarily in cells requiring iron. Its amount in the blood plasma mirrors the number of cells expressing the receptor and its density on cells, making it strongly associated with iron needs and the proliferation rate of erythrocytes [70]. Elevated levels of sTfR indicate insufficient availability of cellular iron, which translates into a clinical outcome [68,71,72].

### 4.4. Hepcidin

Hepcidin is a hormone produced in the liver in response to pro-inflammatory signals (mainly via the interleukin-6 pathway), and its main role is to downregulate iron efflux from enterocytes, hepatocytes and macrophages into the bloodstream. It does so by binding to ferroportin 1 and triggering its lysosomal degradation [64]. Low levels of serum hepcidin indicate depleted iron stores, regardless of anemia [68]. Recent studies have combined low serum hepcidin and high serum sTfR as an alternative mode to diagnose ID in HF. In the study of 165 patients with AHF, this definition has proven hepcidin and sTfR as useful biomarkers for identifying ID (depicting both depleted iron stores and unmet need for iron utilization) regardless of anemia, as well as a predictor of poor outcomes in patients with the most severe ID [68].

### 4.5. Clinical Perspectives

The biomarkers of iron metabolism in HF are a splendid example of molecules that have both strong pathophysiological and clinical meaning that translates into everyday practice. The link between iron metabolism, pathophysiological basics, heart failure symptoms (i.e., exercise tolerance) and the prognosis is a great reflection of the biomarker’s value and utility translated into HF guidelines. Thus, it is strongly recommended that all patients with HF be periodically screened for anemia and ID, and any deviation should be treated as a therapeutic target in both AHF and CHF [2].

## 5. Conclusions

With its complicated pathophysiology, multiple clinical manifestations and long-drawn-out prodromal phase, heart failure needs biomarkers that reflect the wide spectrum of disease, from the earliest manifestations to the terminal stages.

The role of biomarkers in clinical practice is indisputable. Well-known cardiovascular biomarkers such as NT-proBNP or highly sensitive troponins are widely applied as diagnostic tools, popularized by guidelines. These biomarkers represent underlying HF pathophysiology that translates into clinical events and prognosis.

An interesting implementation of biomarkers in clinical practice is biomarker-guided therapy, where clinical management is dependent on biomarker levels. Serial measurements of an established biomarker may give information about clinical response and therapy effectiveness.

Moreover, the evaluation of changes and the relationship among cardiovascular biomarkers are used in numerous ongoing clinical trials, especially regarding heart failure with preserved ejection fraction.

In our review, we summarize the current knowledge in biomarker research related to the development of biomarkers for heart failure. Interestingly, it is possible to study all the abovementioned molecules simultaneously, in line with the multimarker approach. The multimarker approach seems to be a future perspective for cardiovascular biomarkers. Analysis of numerous biomarkers using artificial intelligence may lead to tailored risk stratification and therapy.

## Figures and Tables

**Table 1 jpm-12-00799-t001:** Cardiovascular extracellular matrix biomarkers and their clinical significance.

Biomarker	Clinical Significance
Galectin-3	Prediction of HF development; Correlation with cardiovascular and all-cause mortality
The soluble isoform of suppression of tumorigenicity 2 (sST2)	Prediction of HF development; Correlation with cardiovascular morality
Matrix metalloproteinases (MMP)	Prediction of post-MI HF development; Correlation with post-MI mortality
Matrix metalloproteinases and tissue inhibitors of matrix metalloproteinases balance (MMP/TIMP)	Prediction of HF development
Osteopontin (OPN)	Assessment of HF severity; Association with reverse cardiac remodeling
Interleukin-6 (IL-6)	Prediction of HFpEF development; Association with renal function in HF; Prediction of unscheduled HF hospitalization; Correlation with all-cause, cardiovascular and non-cardiovascular mortality
Syndecan-4 (Syn-4)	Correlation with left ventricle geometrical parameters in CHF
Myostatin (MSTN)	Assessment of HF severity; Prediction of CHF development; Correlation with CHF survival rate and number of rehospitalizations
Procollagen Type-1 C-terminal propeptide (PICP)	Prediction of hypertensive origin HF development; Correlation with ventricular hypertrophy, diastolic function and cardiac fibrosis
Procollagen type III N-terminal propeptide (PIIINP)	Prediction of HFpEF development; Correlation with all-cause and cardiovascular mortality

Abbreviations: HF—Heart Failure, CHF—Chronic Heart Failure, MI—Myocardial Infarction, HFpEF—Heart Failure with preserved Ejection Fraction.

**Table 2 jpm-12-00799-t002:** Clinical application of the assessment of H-FABP in HF.

Acute HF	Chronic HF
Additional value of combining with NT-proBNP to rule out the diagnosis of AHF [48] Elevated serum levels as a predictive marker of acute kidney injury [49] Additional value of either elevated serum levels or in combination with troponin I and NT-proBNP for prognostic assessment [48,49]	In HF with preserved ejection fraction, it is an independent predictor of cardiovascular events [2]

Abbreviations: HF—Heart Failure, AHF—Acute Heart Failure, NTproBNP—N terminal pro B natriuretic peptide.

**Table 3 jpm-12-00799-t003:** Biomarkers of iron deficiency.

Application	Biomarkers
Measurement of functional and storage iron pools	Iron Transferrin Total iron-binding capacity Ferritin Non-transferrin-bound iron Labile plasma iron levels
Assessment of proteins regulating iron absorption and release from tissue stores	Serum hepcidin Soluble hemojuvelin Soluble ferroportin-1
Assessment of proteins regulating the erythropoietic activity of bone marrow	Erythroferrone Soluble transferrin receptor

## Data Availability

Not applicable.

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
