# Peer review of "Biomarkers of Myocardial Injury and Remodeling in Heart Failure"

_jpm, 2022, doi:10.3390/jpm12050799_

Round 1
Reviewer 1 Report
In the current manuscript, Ponikowska et al. present a review on biomarkers of heart failure. The strength of this review is the breadth of biomarker categories covered. Overall, the manuscript is well-written and can serve as useful resource for HF researchers.
Below are my comments:
- The introduction is very brief. It can be expanded to include the types of biomarkers (for example, those used for diagnosis vs. those used to classify severity of disease).
- Discussion of classic HF biomarkers such as NT-proBNP and MR-proANP is missing.
- Table 1 seems to be a repetition of the biomarkers already covered in the text. Perhaps the authors can use the table to mention biomarkers not included in text.
- Conclusions or future directions section is lacking.
Reviewer 2 Report
It is an interesting review about novel biomarkers in heart failure (myocardial injury, remodeling) and in iron metabolism. Although the review refers to a wide spectrum of biomarkers it gives the impression that it is just a simple list of them. In my opinion, a deeper analysis and a more detailed discussion needs to be done. For which of them there is a true necessity for use in every day clinical practice? Is this a realistic target for the near future? Are there ongoing studies about their utility? Will they provide us with more information than the already used ones and will this have an impact in our therapeutic approaches? Moreover the authors have to make clear (and incorporate into the title) their choice to refer also to the iron metabolism. Of course, it is known that iron deficiency is relatively common in patients with heart failure, but there are also other common co-morbidities, like for example renal failure.
Round 2
Reviewer 2 Report
In this revised version the authors sufficiently improved their manuscript providing interesting and adequate details on the topic.